# Spatial clustering of tuning in mouse primary visual cortex

Dario L. Ringach[1,2], Patrick J. Mineault[1], Elaine Tring[1], Nicholas D. Olivas[1], Pablo Garcia-Junco-Clemente[1,†] & Joshua T. Trachtenberg[1]

The primary visual cortex of higher mammals is organized into two-dimensional maps, where the preference of cells for stimulus parameters is arranged regularly on the cortical surface. In contrast, the preference of neurons in the rodent appears to be arranged randomly, in what is termed a salt-and-pepper map. Here we revisited the spatial organization of receptive fields in mouse primary visual cortex by measuring the tuning of pyramidal neurons in the joint orientation and spatial frequency domain. We found that the similarity of tuning decreases as a function of cortical distance, revealing a weak but statistically significant spatial clustering. Clustering was also observed across different cortical depths, consistent with a columnar organization. Thus, the mouse visual cortex is not strictly a salt-and-pepper map. At least on a local scale, it resembles a degraded version of the organization seen in higher mammals, hinting at a possible common origin.

[1] Department of Neurobiology, David Geffen School of Medicine, University of California, Los Angeles, California 90095, USA. [2] Department of Psychology, University of California, Los Angeles, California 90095, USA. † Present address: Instituto de Biomedicina de Sevilla, IBiS, Hospital Universitario Virgen del Rocío/CSIC/Universidad de Sevilla and Departamento de Fisiología Médica y Biofísica, Universidad de Sevilla, and CIBERNED, 41013 Seville, Spain. Correspondence and requests for materials should be addressed to D.L.R. (email: dario@ucla.edu).

A major feature of the primary visual cortex in higher mammals is its organization into two-dimensional feature maps, where the preference of cells for stimulus parameters is mapped on the cortical surface in a quasi-periodic pattern[1–4]. In sharp contrast, the preference of neurons for stimulus attributes in rodents appears to be arranged randomly, in what is termed a salt-and-pepper map[5–7]. The coexistence of sharp tuning at the single cell level[8] with a salt-and-pepper organization has spawned fundamental questions about the origin and significance of maps for visual processing[9–11], the specificity of cortical connections[7,12–17] and the existence of unifying principles of cortical organization across different species[10,11,18–21].

Recent studies have indicated the presence of spatial organization in the primary visual cortex of the rodent that had gone undetected. One demonstrated the anatomical existence of patchy projections from the lateral geniculate nucleus (LGN) and higher cortical areas into layer 1 of V1, and its association with the expression of M2 muscarinic acetylcholine receptor and higher spatial frequency preference at those locations[22]. Another showed that there is a statistically significant clustering of tuning in inhibitory parvalbumin (PV) cells in terms of their orientation preferences[23].

Here we revisited the spatial organization of pyramidal cell-receptive fields in the mouse primary visual cortex by measuring the tuning of neurons in the joint orientation and spatial frequency domain[24,25]. To anticipate the results, we found that the similarity of tuning decreases as a function of cortical distance, with a length constant of ∼40 μm. Thus, despite a substantial diversity of tuning[26], the mouse visual cortex is not strictly salt-and-pepper. In addition, we observed clustering across cortical depths, consistent with a columnar organization[22]. Altogether, the present data suggest that the mouse primary visual cortex is more spatially organized than that previously thought.

## Results

### V1 neurons are tuned for orientation and spatial frequency.

We measured the tuning of pyramidal cells in primary visual cortex by means of resonant, two-photon microcopy in alert, head-fixed mice, expressing GCaMP6f (ref. 27) in the superficial layers of primary visual cortex (V1; Fig. 1a). The visual stimulus consisted of a 20 min-long sequence of flashed, high-contrast, sinusoidal gratings that had random orientations and spatial frequencies (a subset of Hartley basis functions)[24,28] refreshed at a rate of four frames per s (Fig. 1b). Individual cells were segmented from the data and their spiking inferred by subtracting potential contributions by the nearby neuropil and performing non-negative deconvolution of the calcium signals (Fig. 1c, see Methods).

We estimated the tuning of each cell in the joint spatial frequency and orientation domain by linearly regressing the response on the stimulus (Fig. 2a, Methods). This analysis averages the responses of neurons at a given orientation and spatial frequency across spatial phase, thereby generating a tuning profile for both simple and complex cells[25]. Similarly to what is observed in higher mammals, cells in mouse V1 are jointly tuned in orientation and spatial frequency (Fig. 2a).

### Mouse V1 is not strictly a salt-and-pepper map.

To measure the similarity between two tuning profiles, we used their correlation coefficient. The cortical distance between a pair of cells was defined as the distance between the centre of mass of their cell bodies in the image plane, which was parallel to the cortical surface.

The hypothesis of a salt-and-pepper organization predicts the statistical independence of tuning similarity and cortical distance. Instead, the data show that the average similarity decreases as a

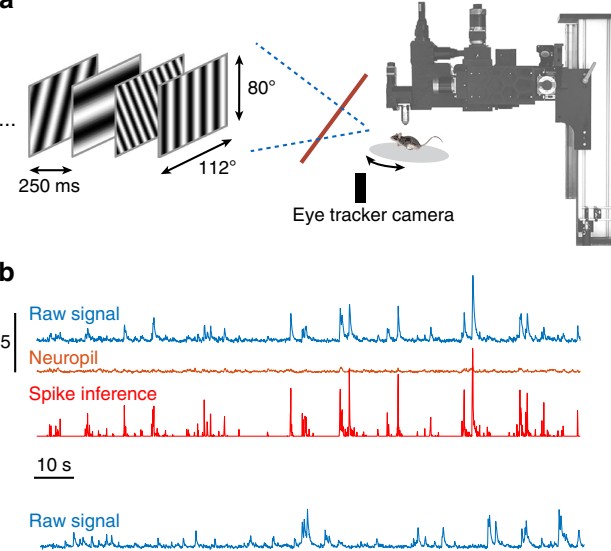

**Figure 1 | Experimental set-up. (a)** Activity of cells in the primary visual cortex was imaged with a two-photon scanning microscope while the mouse observed a continuous visual stimulus on a freely rotating platform. The position of the platform was monitored with an optical rotary encoder. An infrared light-reflective glass (red line) allowed a camera to image the pupil while allowing for unobstructed visual stimulation. The visual stimulus consisted of a sequence of pseudo-random sinusoidal gratings (see Methods for details). **(b)** Two segments illustrating the process of inferring spiking activity from imaging data. First, the calcium fluorescence corresponding to cell bodies (raw signal) and their immediate neighbourhood (neuropil) as a function of time are extracted after compensating for motion in the imaging plane. Here both the raw signal and the neuropil are normalized by the s.d. of the raw signal. The vertical scale bar corresponds to five times the s.d. of the signals. A potential contamination of the signal by the neuropil is ameliorated by projecting out a robust linear prediction of the signal based on the neuropil. Finally, the probability of spiking is inferred by non-negative deconvolution (see Methods for details). The result is a trace that is nearly identical to zero in regions devoid of spiking activity (red trace), further minimizing small contributions of the neuropil to background activity. The spike inference trace is plotted in arbitrary units.

function of cortical distance (Fig. 2b). The median similarity up to distances of 100 μm is significantly higher than that seen at separations of 200 μm (Fig. 2b, values, tailed rank-sum test, indicated by size of data points), and an exponential fit to the data yields a length constant of 38 μm (95% confidence interval (24–50 μm)). A negative correlation between tuning similarity and cortical distance was significant at the 0.05 level in 57% (42 out of 74) of the experiments when tested individually (one such case is shown in Fig. 2c).

The tuning similarity curve asymptotes at a value larger than zero at large cortical distances. This reflects the fact that the average tuning profile across the population is biased, having a prominent response at low spatial frequencies (Fig. 2b, inset).

Tuning similarity, defined by the correlation coefficient between tuning profiles, takes into account the entire shape of the tuning distribution, including the location of the peak and its spread (or bandwidth). We verified that clustering remains present if we compare the distribution of differences in the

preferred orientations for cell pairs at cortical distances less than 50 μm versus pairs more than 150 μm away from each other (Fig. 2d, tailed, rank-sum, $P < 2 \times 10^{-6}$, $n = 1{,}507$ and $12{,}763$). Thus, the observed dependence of tuning similarity with distance is not just a reflection of the clustering of spatial frequency preference reported recently[22], which was also a feature present in our data (tailed, rank-sum test, $P < 0.01$). However, the strongest evidence of clustering of tuning is obtained by analysing the joint selectivity of neurons in the Fourier domain (Fig. 2a,b).

**Tuning similarity clusters along cortical columns.** We also examined the dependence of tuning similarity as a function of distance on the cortical surface and across different depths within layer 2/3 (Fig. 3a). The goal was to investigate a possible clustering of tuning along vertical columns. In four mice, we measured the tuning selectivity of cells in at least four equidistant optical planes parallel to the cortical surface. To ensure independent measurements, we chose a distance of 40 μm between adjacent planes (the support of the point-spread function of the microscope as measured with fluorescent beads was limited to ±8 μm along the z axis). We then computed the dependence of tuning similarity as a function of cortical distance within and across planes (Fig. 3b). We found that tuning similarity was above chance levels even for cells separated by as much as 120 μm in depth (Fig. 3b), consistent with the presence of cortical columns.

**Control experiments.** AAV-syn-GCaMP6 is expressed both in excitatory and inhibitory cells in different degrees[29,30]. Interneurons are known to pool the responses of nearby pyramidal cells inheriting some of their tuning properties[23,31]. Thus, the inclusion of interneurons in our analyses may result in clustering that, perhaps, would not be observed without such contamination. To address this concern we showed that excitatory and inhibitory cells can largely be separated based solely on the statistics of their GcaMP6 signals and that excluding inhibitory cells from our analyses does not affect the main finding.

In four control experiments, we genetically labelled PV-expressing inhibitory interneurons with tdTomato (Fig. 4a). We measured the fluorescent signals originating from pyramidal and PV+ cells during visual stimulation (Fig. 4b). We observed that pyramidal cells show traces with frequent and large and fast calcium spikes, while PV neurons showed slower dynamics and much smaller fluctuations around their mean (Fig. 4b).

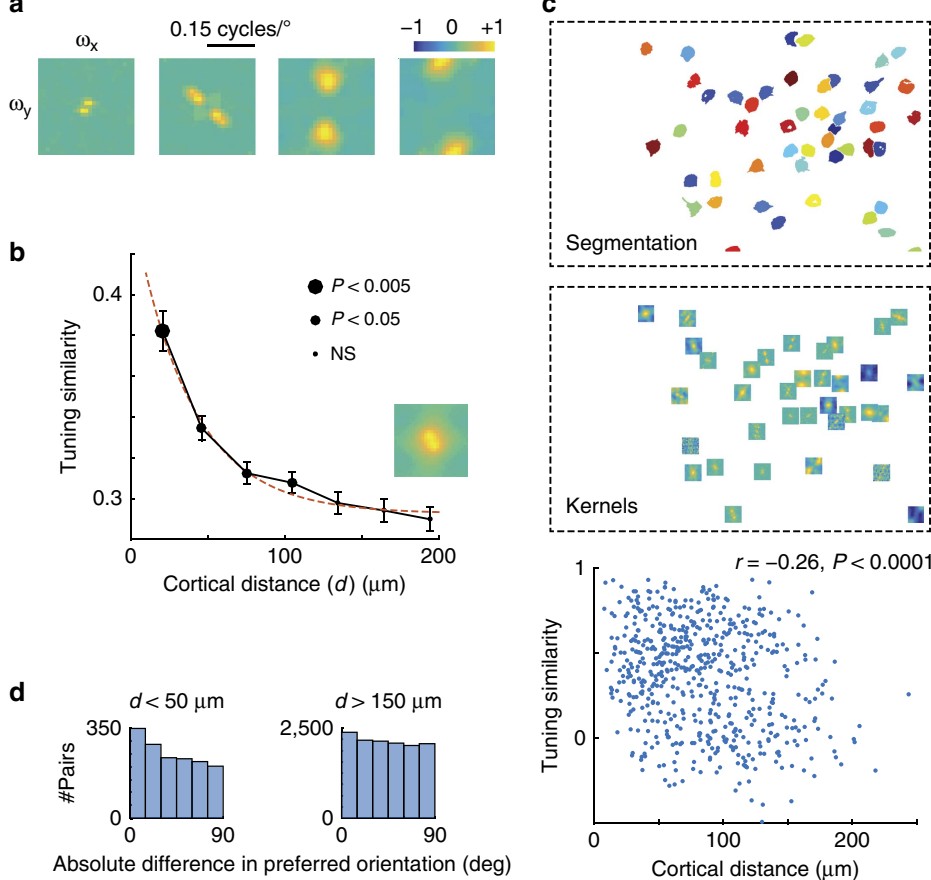

**Figure 2 | Tuning similarity depends on cortical distance in mouse V1.** (**a**) Sample of estimates of the joint tuning of neurons in the orientation and spatial frequency plane in four cells. The origin is in the middle of the image. Kernels are individually normalized, so the maximum absolute value is one. (**b**) Average tuning similarity decreases as a function of cortical distance. Error bars represent ±1 s.e.m. The size of the data points denotes the significance of a rank-sum test comparing the median distribution of data at a given distance to the distribution of the rightmost bin near 200 μm. Red, dashed line shows the best exponential fit to the data. Inset: the mean tuning profile in mouse V1. The number of cell pairs in each group, in order of increasing cortical distance, are $n = 894$, 2,538, 3,473, 3,845, 3,792, 3,399 and 2,964. (**c**) Demonstration of clustering within a single imaging field. Top, segmented cells. Middle, estimated kernels. Bottom, scatter plot of tuning similarity versus cortical distance. There is a statistically significant negative correlation between receptive field similarity and cortical distance. (**d**) Distributions of the absolute difference in preferred orientation as a function of cortical distance for cells within 50 μm of each other (left panel) and at least 150 μm away from each other (right panel) for the data in (**b**).

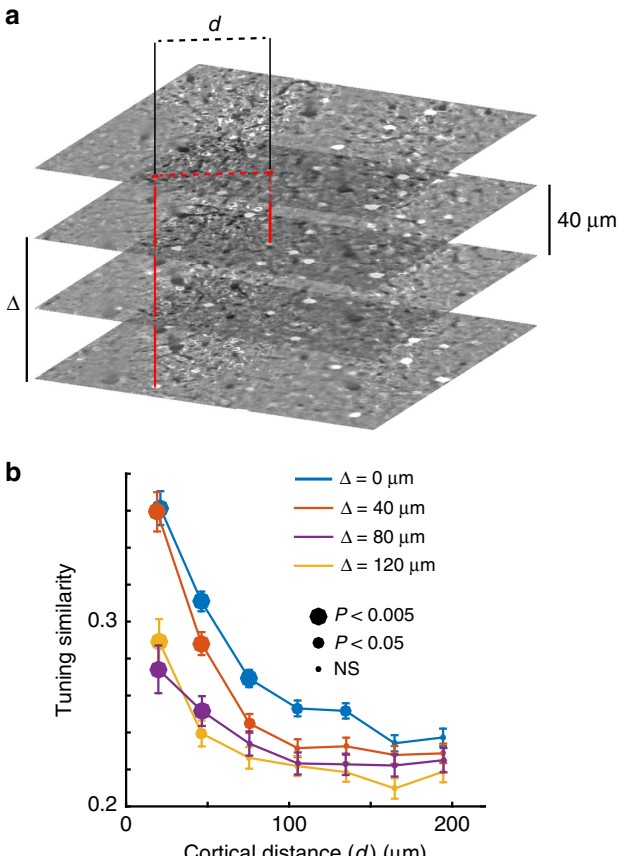

**Figure 3 | Tuning similarity as a function of cortical distance and depth.**
(**a**) The tuning similarity of cell pairs within or across different cortical planes was compared with each other. The cortical distance between two cells, $d$, is defined as the distance between their projections on the cortical surface. Depth difference, $\Delta$, is defined as the difference in depth between the imaging planes. (**b**) Average dependence of tuning similarity as a function of cortical distance and cortical depth. For each individual curve, the size of the data points denotes the significance of a rank-sum test comparing the median distribution of data at a given distance to the distribution of the rightmost bin near 200 μm.

A measure that captures such differences is the kurtosis of the signal distributions. Kurtosis is a measure of how outlier-prone a distribution is. Calcium spikes in pyramidal cells generated signals with high kurtosis, such that 99% of the cases lied above a value of 7.35 (Fig. 4c). In contrast, the mean kurtosis for PV cells was 3.98, and only 1 out of 62 PV + cells (1.6%) had a kurtosis that exceeded the 1% threshold for pyramidal cells (Fig. 4c).

In other experiments we also found similar differences between the kurtosis of SOM (somatostatin expressing neurons, $n = 204$) and VIP neurons (vasoactive intestinal peptide expressing neurons, $n = 357$; Fig. 4d) and that of pyramidal cells. Altogether, these data show that only 1% of inhibitory cells of any type (PV, SOM or VIP) exceeded a kurtosis value of 15. The clustering of tuning similarity remained unaffected by the removal of all cells with a kurtosis value lower than 15 (Fig. 4e). This finding refines a previous result that reported a weak clustering of orientation preferences in PV − cells—a group that might have included contributions from SOM and VIP neurons as well[23].

Interneurons tend to be more broadly tuned for orientation and more low-pass in spatial frequency than pyramidal cells. Thus, an alternative way to bias the analysis towards pyramidal cells is to restrict the data set to neurons whose kernels had sharp tuning. We did this by requiring the peak spatial frequency of the neuron to be larger than 0.025 cycles per degree. Clustering of tuning similarity was still observed after restricting the analysis to sharply tuned kernels (Fig. 5a).

Our semi-automatic cell segmentation procedure involves a human subject confining the defined region-of-interest (ROIs) to cell bodies. Nevertheless, this is a manual process and it is possible that some cell dendritic processes were inadvertently included in the data. One concern is that adding the dendritic branch of a cell that has already been segmented could generate clustering when none was present. Dendritic processes would tend to be represented by ROIs with relatively smaller areas than those defining cell bodies. Thus, one way to remove potential dendritic processes is to exclude small ROIs from the analysis. We verified that clustering of tuning similarity was still observed after removing cells with the smallest 20% ROIs (Fig. 5b).

## Discussion

Our findings show that excitatory cells in layer 2/3 of mouse V1 are functionally clustered into mini-columns[32] that are biased towards a common tuning profile. While it is undeniable that the local diversity of tuning preference in the rodent is much higher than that found in carnivores and primates, it is equally clear from our results and that of others[22,23] that the notion that mouse visual cortex is organized as a strict, salt-and-pepper map ought to be rejected (Fig. 2). This unexpected finding prompted us to examine in more detail some of the published data in an effort to understand the reasons for the apparent discrepancy with previous reports.

An influential study that first investigated the micro-organization of rat visual cortex using two-photon imaging reported no significant relationship between the direction preference of cells and their cortical distance[5]. However, a closer look at their data (Fig. 6a in ref. 5) reveals that cells within 50 μm of each other tend to share the same direction preferences (Fig. 6a). Indeed, a re-analysis of these data, limited to cell pairs at most 100 μm from each other, shows a statistically significant dependence of relative preferred direction on cortical distance ($P < 0.003$, $r = 0.22$; Fig. 6a). We suspected that a failure to find a significant relationship in the original study was because of the inclusion of data points with cortical distances between 100 μm and up to 250 μm, comprising a large number of cell pairs with disparate direction preferences. Consistent with their report, the correlation between relative direction preference and cortical distance in this larger data set is not significant ($P > 0.1$, $r = 0.07$). The reason is that the additional data points are masking an existing dependency that occurs within a narrower range. An analysis closer to our approach (Fig. 2d) consists of comparing the distributions of relative, preferred direction computed at small and large cortical distances. Indeed, such analysis reveals that nearby cells have a smaller difference than those that are farther away (tailed rank-sum test, $P < 0.02$, Fig. 6b).

Another study analysed the similarity of linear receptive fields of simple cells in layers 2/3 and emphasized the large diversity of the local population[26]. Nonetheless, as pointed out by the authors, their data showed traces of spatial clustering, with cells within 100 μm of each other having receptive fields that were more similar than those 100–200 μm apart (see Fig. 6 in ref. 26).

The large, local diversity of receptive fields in rodent V1 was also evident in earlier studies that used single-electrode recordings to study receptive field properties, but the limited spatial sampling obtained in these experiments make the data inadequate to rule out the hypothesis of a salt-and-pepper organization[33,34].

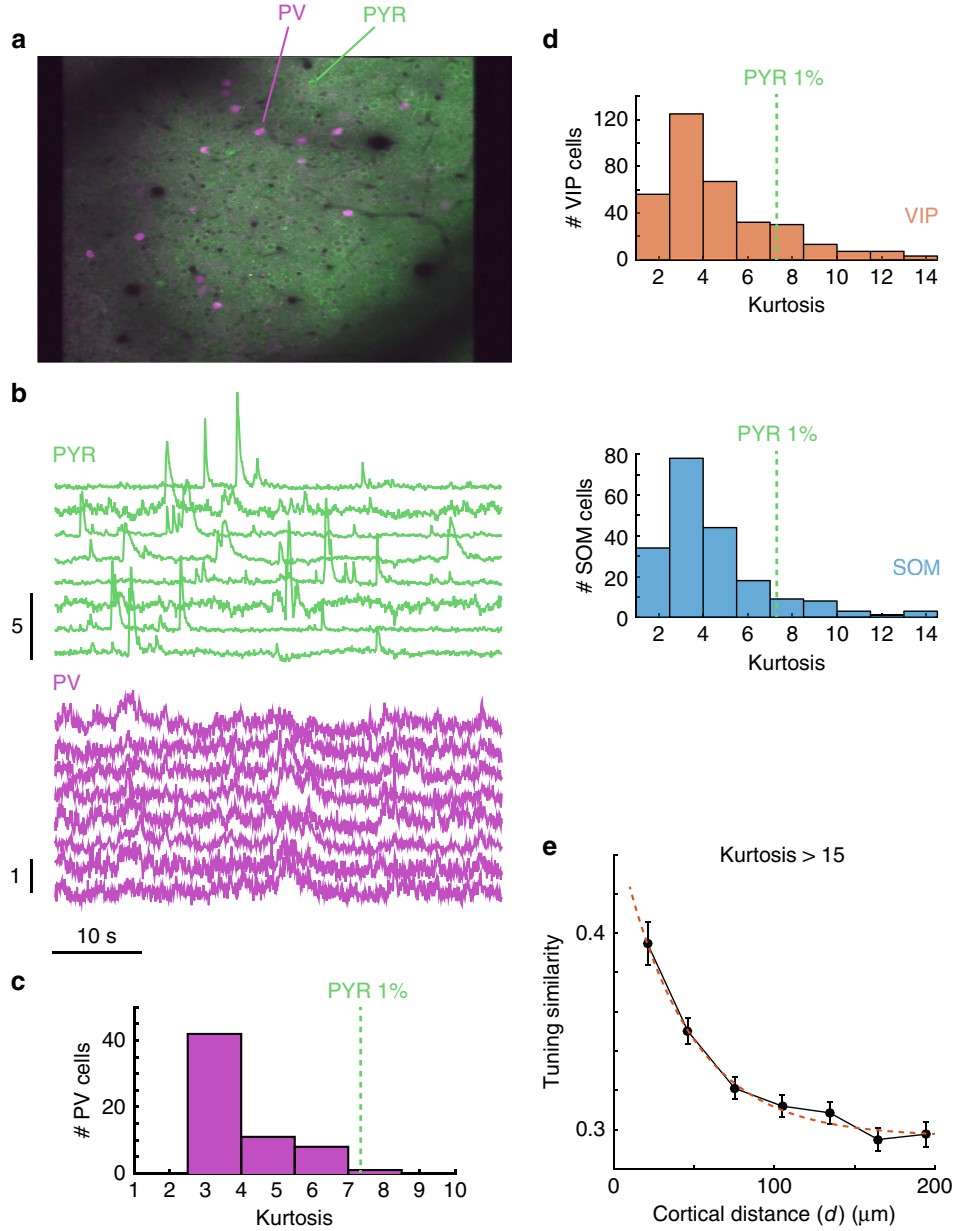

**Figure 4 | Inhibitory and excitatory calcium signals have different distributions.** (**a**) Two-photon image of a preparation where PV cells are genetically labelled with tdTomato. (**b**) Sample traces for PV + and (putative) pyramidal cells. The signals are qualitatively different. (**c**) The kurtosis of the signal distributions differs strongly between PV cells (shown) and that of excitatory neurons (not shown). Only 1% of the putative excitatory cells have kurtosis values lying to the left of the dashed vertical line. (**d**) Distribution of kurtosis for VIP and SOM cells has also low kurtosis values. (**e**) Clustering of tuning similarity remains largely unmodified even after removing all cells with kurtosis values less than 15.

The present findings are consistent with the recent demonstration of patchy projections from the LGN and higher cortical areas into layer 1 of V1, and its association with the expression of M2 muscarinic acetylcholine receptor and higher spatial frequency preference at those locations[22]. The reported clustering of orientation tuning preference in PV cells is consistent with the present findings[23]. This is because PV neurons pool the responses of nearby pyramidal cells and the resulting bias in their tuning may reflect local homogeneities in the tuning of pyramidal cells which, according to the present data, show a degree of clustering. Altogether, growing evidence suggests that mouse V1 may be more spatially organized than previously thought.

It is worth emphasizing that the degree of spatial clustering in the mouse is weak compared with that observed in higher mammals[5,35]. Its presence, however, suggests that a similar mechanism underlying the generation of cortical columns may be at play in all mammalian species. Given the spatial scales involved (Fig. 2), it is possible that the spatial clustering results from the anatomical development of cortical minicolumns, where radially arranged sister neurons[36] acquire similar tuning and connections with each other[14,16,37,38]. Such anatomical, minicolumns are shared by all mammalian species, including monotremes, and are postulated to have evolved in mammalian ancestors alongside the emergence of a six-layered cortex[39]. In this context, one hypothesis that deserves further study is the idea that the

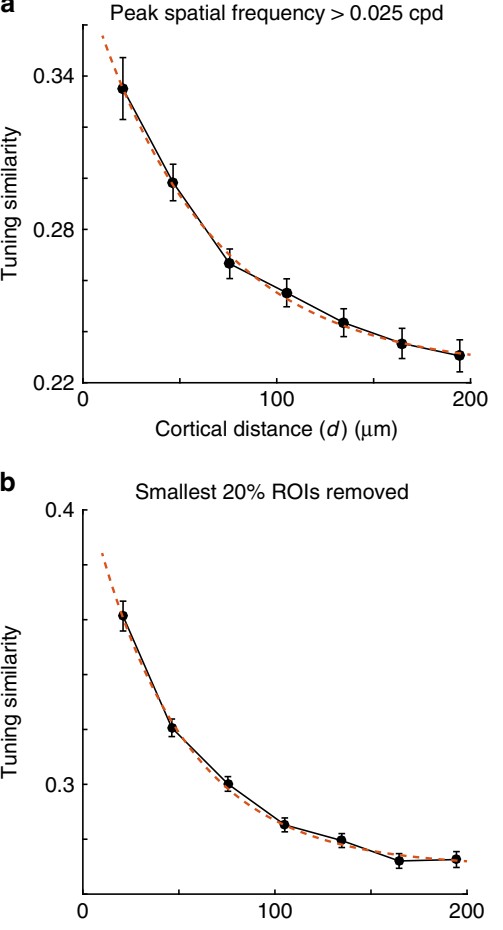

**Figure 5 | Robustness of clustering.** (**a**) Clustering of tuning similarity remains unaffected if the analysis is restricted only to cells with sharp tuning selectivity, defined here as kernels with peak spatial frequency at values larger than 0.025 cycles per degree. (**b**) Clustering of tuning similarity remains unaffected after removing the ROIs with the smallest 20% areas, thereby ruling out that the occasional inclusion of dendritic processes may be responsible for the results.

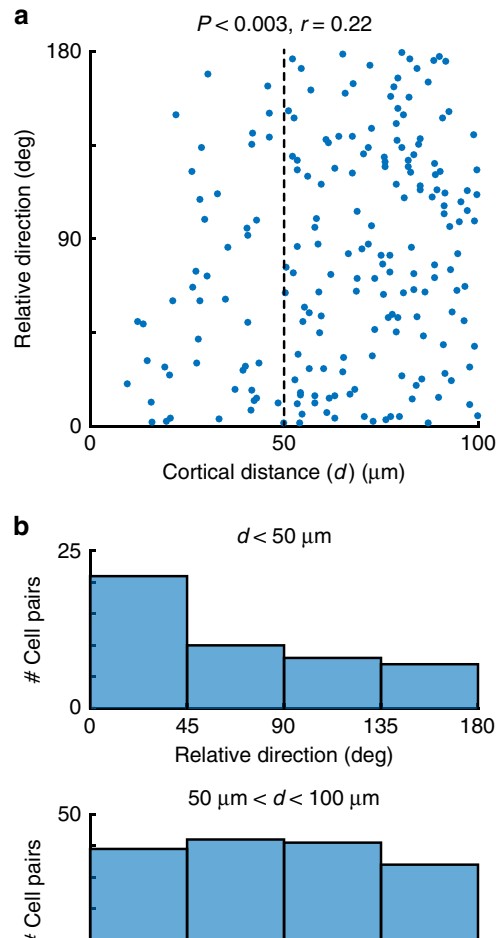

**Figure 6 | Re-analysis of data from a seminal study[5] shows evidence of spatial clustering.** (**a**) Dependence of relative preferred direction as a function of cortical distance restricted to cells that are no farther apart than 100 μm. Data are replotted from Fig. 6a of the original study. There is a statistical dependence of relative direction with cortical distance ($P < 0.003$, $r = 0.22$). (**b**) The distribution of relative directions for cells within 100 μm of each other is statistically different from that of cells distanced between 50 and 100 μm away (tailed rank-sum test, $P < 0.02$).

degree of spatial clustering in tuning properties across species depends on differences in the horizontal dispersion of clonally related cells, which appear to be larger in mouse than in primate[38]. Such a mechanism predicts, and is consistent with, the larger retinotopic scatter of receptive fields in species showcasing a larger diversity in the local population[40,41].

The finding that tuning is spatially clustered in mouse V1 raises a natural question: do mice have feature-based cortical maps? In other words, is there a detectable, regular tiling of preferences as seen in higher mammals? So far we have not seen any evidence this is the case, but our data are limited by small fields of view because of the restricted expression of GCaMP6 with adeno-associated virus (AAV) delivery. However, the quasi-periodic pattern of spatiotemporal sensitivity described recently[22] suggests that this might be a real possibility. We are currently conducting studies using transgenic animals and a larger field of view that would allow us to test this tantalizing possibility in the near future.

## Methods

**Animals.** All procedures were approved by University of California, Los Angeles (UCLA's) Office of Animal Research Oversight (the Institutional Animal Care and

Use Committee), and were in accord with the guidelines set by the US National Institutes of Health. A total of 30 C57BL/6J mice (Jackson Laboratory), both male (10) and female (20), aged P35–56, were used in this study. Mice were housed in groups of two to three in reversed light cycle. Animals were naive subjects with no prior history of participation in research studies. We imaged 129 different fields, and obtained data for 7,018 cells, for a median of 47 cells per field (range: 6–162).

**Surgery.** Carprofen and buprenorphine analgesia were administered preoperatively. Mice were then anaesthetized with isoflurane (4–5% induction; 1.5–2% surgery). Core body temperature was maintained at 37.5 °C using a feedback heating system. Eyes were coated with a thin layer of ophthalmic ointment to prevent desiccation. Anaesthetized mice were mounted in a stereotaxic apparatus. Blunt ear bars were placed in the external auditory meatus to immobilize the head. A portion of the scalp overlying the two hemispheres of the cortex (~8 mm by 6 mm) was then removed to expose the underlying skull.

After the skull is exposed it was dried and covered by a thin layer of Vetbond. After the Vetbond dries (~15 min) it provides a stable and solid surface to affix an aluminium bracket with dental acrylic. The bracket is then affixed to the skull and the margins sealed with Vetbond and dental acrylic to prevent infections.

**Virus injection.** A 3 mm diameter region of the skull overlying the occipital cortex was removed. Care was taken to leave the dura intact. GCaMP6-fast (UPenn Vector Core: AAV1.Syn.GCaMP6f.WPRE.SV40; #AV-1-PV2822) was expressed in cortical neurons using AAV. AAV-GCaMP6-fast (titre: $\sim$13 genomes ml$^{-1}$) was loaded into a glass micropipette and slowly inserted into the primary visual cortex (V1) using a micromanipulator. Two injection sites were centred around the centre of V1 and separated about 200 μm apart. For each site, AAV-GCaMP6-fast was pressure-injected using a PicoSpritzer III (Parker, Hollis, NH; four puffs at 15–20 pounds per square inch with a duration of 10 ms; each puff was separated by 4 s) starting at a depth of 350 μm below the pial surface and making injections every 10 μm moving up with the last injection made at 100 μm below the pial surface. The total volume injected across all depths was $\sim$0.5 μl. The injections were made by a computer programme in control of the micromanipulator and the Picosprtizer.

A sterile 3 mm diameter cover glass was then placed directly on the dura and sealed at its edges with VetBond. When dry, the edges of the cover glass were further sealed with dental acrylic. At the end of the surgery, all exposed skull and wound margins were sealed with VetBond and dental acrylic and a small, sealed glass window was left in place over the occipital cortex. Mice were then removed from the stereotaxic apparatus, given a subcutaneous bolus of warm sterile saline, and allowed to recover on the heating pad. When fully alert they were placed back in their home cage.

**Control experiments.** To identify signals originating from PV-expressing neurons, we genetically labelled them. PV-IRES-Cre knock-in female mice (Jackson Laboratories, stock no. 008069, generated by S. Arbor, FMI) were crossed with male tdTomato reporter knock-in mice directly received from Jackson Laboratory (stock no. 007905, 'Ai9', generated by H. Zeng, Allen Brain Institute). All experimental mice were hemizygous for both transgenes (PV-Cre:Ai9). Homozygous PV-IRES-cre mice used for the above breeding were from a F1 cross of a male and female directly received from Jackson Laboratory.

We conducted similar experiments to isolate signals originating from VIP and SOM inhibitory neurons. Homozygous VIP-IRES-Cre mice (Jackson Laboratories, stock no. 010908, generated by Z. Josh Huang, CSHL) were crossed with homozygous tdTomato reporter knock-in mice (Jackson Laboratories, stock no. 007905, 'Ai9', generated by H. Zeng, Allen Brain Institute). All mice were hemizygous for both transgenes (VIP-Cre:Ai9 HET). VIP-Cre HET/Ai9 HET was then backcrossed with parental homozygous VIP-IRES-Cre mice from JAX; therefore, 25% offspring were homozygous for VIP-Cre and hemizygous for tdTomato. Homozygous SOM-IRES-Cre mice (Jackson Laboratories, stock no. 013044, generated by Z. Josh Huang, CSHL) were crossed with homozygous tdTomato reporter knock-in mice (Jackson Laboratories, stock no. 007905, 'Ai9', generated by H. Zeng, Allen Brain Institute). All mice were hemizygous for both transgenes (SOM-Cre:Ai9 HET). SOM-Cre HET/Ai9 HET was then backcrossed with parental homozygous SOM-IRES-Cre mice from JAX; therefore, 25% offspring were homozygous for SOM-Cre and hemizygous for tdTomato. Finally, we crossed VIP-Cre HOM/Ai9 HET with SOM-Cre HOM/Ai9 HET mice; therefore, 50% offspring were hemizygous for the three transgenes (VIP-Cre:SOM-Cre:Ai9 HET). GCAMP6s was expressed in cortical neurons using a Cre-dependent AAV injection (AAV1.Syn.Flex.GCaMP6s.WPRE.SV40; # AV-1-PV2821; titre: $\sim$13 genomes ml$^{-1}$); therefore, just VIP and SOM interneurons expressed the GCAMP6s calcium sensor.

**Imaging.** Once expression of Gcamp6f was observed in primary visual cortex, typically between 11 and 15 days after the injection, imaging sessions took place. Imaging was performed using a resonant, two-photon microscope (Neurolabware, Los Angeles, CA) controlled by the Scanbox acquisition software (Scanbox, Los Angeles, CA). The light source was a Coherent Chameleon Ultra II laser (Coherent Inc, Santa Clara, CA) running at 920 nm. The objective was an ×16 water immersion lens (Nikon, 0.8 numerical aperture, 3 mm working distance). The microscope frame rate was 15.6 Hz (512 lines with a resonant mirror at 8 kHz). Eye movements and pupil size were recorded via a Dalsa Genie M1280 camera (Teledyne Dalsa, Ontario, Canada) fitted with a 740 nm long-pass filter that looked at the eye indirectly through the reflection of an infrared-reflecting glass (Fig. 1a). Images were captured at an average depth of 210 μm (90% of imaging fields within the range 80–320 μm). During imaging a substantial amount of light exits from the brain through the pupil. Thus, no additional illumination was required to image the pupil. The platform was mounted on a rotary, optical encoder (US Digital, Vancouver, WA) connected to an Arduino Mega 2560 board, which provided direct access to movement information. Both locomotion and eye movement data were synchronized to the microscope frames.

**Visual stimulation.** Hartley stimuli[24,28] were generated in real-time with a Processing sketch using OpenGL shaders (see http://processing.org). The stimulus was updated four times a second on a BenQ XL2720Z screen refreshed at 60 Hz. The screen measured 60 cm × 34 cm and was viewed at 20 cm distance, subtending 112 × 80 degrees of visual angle. The maximum spatial frequency was 0.15 cycles per degree. A transistor-transistor logic (TTL) pulse was generated with an Arduino at each stimulus transition. The pulse was sampled with the microscope and time-stamped with the frame and line number that was being scanned at that

time. The time stamps provided the synchronization between visual stimulation and imaging data.

The screen was calibrated using a Photo-Research (Chatsworth, CA) PR-650 spectro-radiometer, and the result used to generate the appropriate gamma corrections for the red, green and blue components via an nVidia Quadro K4000 graphics card. The contrast of the stimulus was 99%. The centre of the monitor was positioned with the centre of the receptive field population for the eye contralateral to the cortical hemisphere under consideration. The location of the receptive fields was estimated by an automated process where flickering checkerboard patches (patch size 12 × 12 deg, checker size 4 deg) appeared at randomized locations within the screen. This experiment was ran at the beginning of each imaging session to ensure the centring of receptive fields on the monitor.

**Motion stabilization.** Calcium images were aligned to correct for motion artefacts in a two-step process. First, we aligned images rigidly in a recursive manner to correct for slow drifts in the imaging plane. Pairs of neighbouring images in time were aligned by finding the peak of their cross-correlation; then, pairs of averages of such pairs were aligned; and so on and so forth. In the second step, we aligned images non-rigidly to a reference mean image to correct for fast in-plane movements, which are frequently observed during grooming. We iteratively applied the Lucas–Kanade algorithm[42] to non-rigidly match a reference mean image, refining the estimate of this reference mean image after each alignment iteration.

**Segmentation.** Following motion stabilization, we used a Matlab graphical user interface tool developed in our laboratory to manually define regions of interest corresponding to putative cell bodies. We used correlation and kurtosis images to identify cell candidates[43]. The correlation image, corresponding to the average correlation of a pixel and its eight neighbours across time, highlights regions of space that covary in time. These images were computed after subtracting linear de-trending[44]. The kurtosis image highlights regions in space with signals composed of large, infrequent deviations or putative spikes, which biased our selection of ROIs towards pyramidal cells.

We used these images to visually identify approximately circular regions of space of an appropriate radius with high correlation, and high kurtosis. Clicking a seed pixel at the centre of such a candidate patch allowed the definition of a ROI by flood-filling an image corresponding to the correlation of the highlighted pixel and every other pixel in the image field[45]. The interface then allowed the user to dynamically grow or shrink the ROI to a desired size.

**Signal extraction and spike inference.** Following segmentation, we extracted signals by computing the mean of the calcium fluorescence within each ROI and discounting the signals from the nearby neuropil. We then used non-negative deconvolution[44,46] to estimate spikes from calcium traces. We solved the inverse, constrained form of the non-negative deconvolution problem[44] using the CVX package[47]. To mitigate the effect of drifting background fluorescence, we modelled the offset as slowly moving in time with a 10-knot cubic spline. We estimated the noise of the measured calcium signals as the median absolute deviation of the first-order derivative divided by a factor of $0.6745\sqrt{2} \approx 0.954$ (ref. 48).

The constrained deconvolution method of Pnevmatikakis et al.[44] requires the specification of the impulse response of the calcium indicator. We assumed an exponential impulse response function and estimated its decay time using reference data consisting of simultaneous loose-seal cell-attached recordings and calcium imaging of GCamp6 in visual neurons[27]. We re-sampled this data set at a sampling rate of 15.5 Hz, ran non-negative deconvolution for a grid of values of decay times, computed the $R^2$ of the estimated calcium signal and the ground-truth cell spike trains across the 11 cells of the data set, and selected the parameter set with the largest mean $R^2$. This yielded a decay time $\tau_{1/2} = 135$ ms for a validated mean $R^2$ of .42 (compare Supplementary Table 3 in ref. 27).

**Linear model.** The response of a neuron to a stimulus was assumed to be given by the linear model:

$$y(t) = \sum_{\omega_x, \omega_y, \tau} s(\omega_x, \omega_y, t - \tau) w(\omega_x, \omega_y) v(\tau) + ar(t) + b + \epsilon(t)$$

Here $\epsilon(t)$ is independent and identically distributed (i.i.d) Gaussian noise, $s(\omega_x, \omega_y, t - \tau)$ is the stimulus presented at time $t$, $w(\omega_x, \omega_y)$ is the Fourier kernel, $v(\tau)$ is the temporal kernel, $b$ is the offset during rest, $a$ is the change in offset during locomotion, $y(t)$ is the measured response and $r(t)$ is an indicator variable taking the value 1 when the instantaneous velocity of the animal is at least $1 \text{ cm s}^{-1}$, and zero otherwise.

We fit this model through alternating least squares[49]. We constrained the norm of the temporal kernel to 1. We used a smoothness penalty for the spatial kernel[50]; its strength was determined by fivefold cross-validation.

We compared the quality of fit of this generic linear model to a baseline model:

$$y(t) = ar(t) + b + \epsilon(t)$$

We considered a fit significant whenever the cross-validated sum-of-squared error

SSE$_L$ of the generic linear model was:

$$r_L = \sqrt{1 - SSE_L/SSE_B} \geq 0.15$$

Here SSE$_B$ corresponds to the sum-of-squared error of the baseline model and $r_L$ is analogous to a correlation (Pearson's $r$) value. In all, 3,803/7,018 were significantly tuned according to this criterion.

**Data selection.** For any one experiment, we accepted tuning profiles of each where the linear model accounted for $r_L \geq 0.15$ (which occurred in ~54% of the population) and the total number of such cells within the field exceeded 20. This resulted in a subset of 45 experiments with a mean of 42 cells (range 20–94) that were analysed here.

**Data availability.** All analyses were conducted in Matlab (Mathworks, Natick, MA). The code and data are available upon request from the authors.

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

## Acknowledgements

We thank Robert Shapley for comments on an earlier version of this manuscript and anonymous reviewers for their constructive feedback. This work was supported by NIH-EY-18322 (D.L.R.) and NIH-EY-23871 (J.T.T.).

## Author contributions

D.L.R. designed research. E.T., N.D.O. and P.G.-J.-C. performed surgeries. E.T., N.D.O., P.G.-J.-C., P.J.M and D.L.R. performed data collection. P.J.M and D.L.R. analysed the data, J.T.T. and D.L.R. wrote the manuscript.

## Additional information

**Competing financial interests:** The authors declare no competing financial interests.

**How to cite this article**: Ringach, D. L. *et al.* Spatial clustering of tuning in mouse primary visual cortex. *Nat. Commun.* 7:12270 doi: 10.1038/ncomms12270 (2016).

