## [Peer Review File · Nature Communications]

Editorial Note: this manuscript has been previously reviewed at another journal that is not operating a transparent peer review scheme. This document only contains reviewer comments and rebuttal letters for versions considered at *Nature Communications*.

Reviewers' comments:

Reviewer #1 (Remarks to the Author):

The authors have addressed all of my concerns.

I have just two minor suggestions for further edits:

Delete "from": "Individual cells were segmented from the data (Fig 1c) and their spiking inferred from by non-negative deconvolution of the calcium signals..."

"The finding that tuning is spatially clustered in mouse V1 raises a natural question -do mice have cortical maps?" This statement should be qualified. There is no doubt that mice have strong retinotopic maps. Perhaps change to "-do mice have feature-based cortical maps?"

Reviewer #2 (Remarks to the Author):

The authors have substantially improved the text and description of their data. This is an important study that will have a large impact on the field. I have only a few comments that I believe will help others realize the importance of this work.

Issues:

- 1) It would be helpful to insert supplementary figure 1 into the manuscript. It is an important figure and it would be much better if supplementary figure 1 were placed between the current figures 2 and 3.
- 2) The procedure for extracting spikes from the Ca^{++} signals is an important methodological consideration. Supplementary figure 2 shows the calcium waveforms, but it would be nice to show the extracted spikes as well. I do not find the extracted spike histogram in figure 1d as very informative as we do not know the underlying calcium signals that they are based upon.
- 3) If no spike extraction were performed, does the same relationship between similarity and distance hold?

4) Why is there a different asymptote in figure 2 and figure 3?

5) The depth relationship shown in figure 3 is very interesting. I wonder if it might be helpful to add a panel that separates the depths by color, but with an x-axis that reflects the actual distance of between the cell bodies instead of just the cortical distance. If depth distances are the same as lateral distances then all the curves should overlap. I think this might strengthen the columnar argument.

Reviewer #3 (Remarks to the Author):

The manuscript is improved from the original submission to Nature, and the new data is useful. Several concerns remain which, if addressed thoroughly, would strengthen the manuscript and its impact on the field.

Previous comments and rebuttal are below, with new comments beginning with --

The authors investigated whether, in mouse primary visual cortex, there is evidence for spatial clustering of neurons that share similar visual orientation and/or spatial frequency tuning. The premise is that previous studies claim that, in contrast to many carnivores and primates, mice and rats lack any such spatial organization in visual cortex. The manuscript is well written. Advances in this manuscript include use of state-of-the-art two-photon microscope, novel methods for eyetracking, novel means for efficient stimulus delivery and mapping, novel sophisticated analyses for brain motion correction and signal extraction. These methods allowed for a high yield of neurons with robust multidimensional tuning curves. However, several considerations limit enthusiasm for publishing this work in Nature: 1) The level of clustering reported is nevertheless quite weak. Whether or not the organization is entirely salt-and-pepper, a change in tuning similarity from .3 to .38 (Fig. 3) for neurons <50 um apart is quite small, and far from many people's conception of a 'mini-column', the term suggested by the authors. See our reply to Reviewer #1, point #2. We agree that the degree of clustering is much smaller than seen in higher mammals and we have modified the text to make this clear. The term "minicolumn" was adopted from the Adams and Horton's review article (which we cite), but the term itself appears to go back to Lorente de No.

This modest level of clustering, if real, may not substantially change other scientists' views of the similarity or differences visual cortex in rodents vs., say, cats, where orientation clustering is qualitatively more homogeneous.

The data allow us to reject a strict salt-and-pepper organization, and we think the result is highly relevant to the ongoing discussion of cortical organization in different species. Though the degree clustering is weaker than seen in higher mammals, the fact that there is a demonstrable effect opens the possibility that similar mechanisms are at play in different species.

---It is unclear why the authors still do not cite and discuss Runyan and Sur, 2013 (mentioned below in initial comments). This paper states:

"The orientation scatter surrounding both RFP_ and RFP_ neurons increased with distance, being significantly greater at 240 um than at 60 or 90 um ($p < 0.05$ comparing scatter of RFP_ or RFP-neurons at 60 and 90 um with 240 um), reflecting the existence of local homogeneities in the

orientation map that dissipate at larger distance scales."

---It appears that, if the goal of the current paper is to reject strict salt and pepper organization, that the above finding in RFP-neurons may already begin to do that. However, if concerns regarding contamination (see below) are addressed, the current data would nevertheless be quite helpful in consolidating and better quantifying the above finding, as the current methods are more efficient and include more neurons from many more experiments.

2) Inhibitory interneurons could account for the majority of the effect observed: The authors did not discriminate excitatory from inhibitory neurons. Further, the authors used AAV1-synGCaMP6. AAV1-syn-XX is known to express in inhibitory neurons as well as excitatory neurons, and to show a bias towards PV+ inhibitory neurons (Callaway and colleagues). Given that inhibitory neurons show broad tuning but still often possess orientation and spatial frequency preferences, it's likely that at least 15-20% of neurons included are inhibitory neurons, with most of these being PV+ interneurons. Many studies have shown that interneurons likely pool the responses of nearby excitatory neurons. Recently, Runyan and Sur (2013, *J. Neurosci.*, Fig. 5E) showed that the orientation preference of PV+ interneurons is more similar to the preferences of non-PV neurons (mostly excitatory) located <60 μm away from the PV+ neuron cell body, and that this similarity in preference is weaker when considering neurons <240 μm away. From this, one can infer that the PV+ neurons are more likely than chance to match the tuning of their nearest neighbors, in which case lumping PV+ and PV-neurons in the current study could very well help explain the modest local clustering result in the current manuscript. This same argument can be applied to the data the authors reanalyzed (current manuscript, Fig. 4) from a previous paper (Ohki et al), which also did not distinguish interneurons from excitatory neurons. Runyan and Sur also show a very weak local clustering of orientation preference (<5%) amongst PV-putative excitatory neurons. This can either be seen as prior evidence that even excitatory neurons have already been shown to exhibit some local clustering. Alternatively, these data could be seen as prior evidence that other studies find only trivially low levels of orientation clustering when excluding PV+ neurons. Similar to Runyan and Sur, a subsequent study also showed evidence of a bias in tuning (in this case, in ocular dominance tuning as well as disparity tuning) of PV+ neurons to nearby PV-neuron tuning, at the 50 μm scale in mouse visual cortex (Scholl .. Zemelman, *Neuron* 2014, Fig. 4). Further, Scholl et al. saw little or no spatial clustering for these other features in mouse V1 when considering PV-neurons (mostly excitatory), further suggesting that V1 in rodent and some carnivores/primates may be qualitatively different in this regard.

This is a key point that the Nature editor asked us to consider as well before resubmission. We now include new data quantifying the level of contamination expected from PV+ neurons; GCaMP6f signals simultaneously measured from pyramidal neurons and PV+ neurons, the latter identified by their expression of tdTomato (n=4 mice), and compared their signal statistics. The signals generated by PV+ cells are easily distinguished from those generated from pyramidal cells, and easily separated by the kurtosis of the responses, as is clear in our new Supplementary Figure 2.

Pyramidal neurons generate large and transient calcium spikes that result in very large kurtosis values (Supplementary Fig 2b, top). In contrast PV+ neurons generate kurtosis values that are only slightly above 3 (corresponding to a Gaussian distribution) (Supplementary Fig 2b, bottom and Supplementary Fig 2c).

Moreover, the density of PV+ cells is much lower than those of pyramidal neurons. Even if we overestimate the relative ratio of PV+:PYR at 1:4, this makes the probability of inclusion of a PV+ cell

in our dataset at $p=0.0025$. Then, the probability that a pair of cells selected in the analysis will include at least one PV cell, will be 0.5% at the most. As additional confirmation, a re-analysis of the data where we reject any cell with kurtosis < 10 (a threshold never exceeded by a PV+ cell in the control experiments) yields effectively the same results reported here.

---The discussion in Supp fig 2 is quite useful, though it remains unclear from the main Methods what criteria were used. What does 'high kurtosis' mean? Can you please define exactly what the kurtosis cutoff is? Was it $k > 7.35$? This is discussed in text in terms of back-of-the-envelope calculations, but not specified in methods as the cutoff that was actually used.

---While the kurtosis arguments are useful, the estimate of 1:4 PV cells doesn't fully address the issue of other GABAergic interneurons, all of which are broadly tuned and should at least be discussed.

---Further, due to overall broader tuning, PV cells are more likely to be driven by one of the chosen SFs and TFs and orientations and stimulus sizes used than a typical non-PV cell, thus the driven subset of cells in the sample will likely be enriched for PV cells, in cases where comprehensive mapping of the entire stimulus space is not performed. These considerations, while unlikely to strongly affect the outcome, merit further discussion and more detailed methods.

---Surprisingly, the Runyan and Sur 2013 paper showing weak clustering among PV-neurons is still not included. A citation to Runyan 2010 is provided at the start of the new section, which would appear to be the wrong citation.

---Also, Scholl / Zemelman reference is still not included, even though this work directly discusses the issue of clustering in PV+ and in PV-neurons in mouse V1.

3) Several technical issues make it difficult to properly evaluate the level of clustering, given the reagents used and the current explanation of the analysis methods. First, the authors use GCaMP6, which is known to drive strong responses in apical dendritic branches. Given the imaging objective used and any residual brain motion, if even a few of these branches were included as spurious cell bodies, they could contribute to local clustering. Nearby proximal dendritic branches from the same neuron will likely have similar tuning and thus could bias the clustering effect, even across planes 40-120 μm apart, since pyramidal cells are radially organized.

As described in the Methods section images were stabilized before signals were extracted. The non-rigid alignment process results in residual motion that is a small fraction of the ROI representing the cell body, and mitigates artifacts that result from motion in the (x,y) plane. Typical examples of such compensation in our setup and how they compare to rigid alignment, together with the code we use, can be seen here:

<http://xcorr.net/2014/08/02/non-rigid-deformation-for-calcium-imaging-frame-alignment/>

---Response to point 3 addresses motion correction in x-y (not in z).

---This response is still incomplete, and doesn't address possibility of multiple dendritic branches being counted. Are results weaker if the smallest diameter ROIs (e.g. possible dendrites from same neuron) are excluded?

A second challenge in parsing the data is the unconventional, albeit interesting, data analysis methods used. Dense infection with GCaMP6 is known to result in huge neuropil signals (Chen .. Svoboda). If not removed perfectly for each neuron, these signals will contaminate the cell body of interest. Neuropil itself can exhibit local tuning biases, further compounding the issues. Here, there is no way to assess if the authors ensured that neuropil contamination was fully eliminated -an important prerequisite for these particular claims. This is even more problematic given the large point-spread function of the imaging objective employed. The contributions from the neuropil were subtracted by estimating a robust linear predictor of the signal based on the surrounding neuropil (excluding other nearby cells) and then subtracting it from the ROI signal. The scatterplot of neuropil vs cell's ROI signal typically shows that the baseline of the cell body ROI is linearly related to the neuropil (both signals z-scored):

---Regarding neuropil -even if the fit removed 90% of neuropil contamination, the residual 10% might be enough to drive the very weak signals shown, particularly in poorly tuned neurons. Neuropil contamination is a serious issue with GCaMP6, it should be taken seriously, particularly when analyzing very subtle effects. Were the clustering finding to be true despite residual neuropil contamination, neuropil contamination would still lead to an overestimate of this already small effect.

---Additional analyses could include comparisons of ori preference of neuropil vs cells, reanalysis of Fig 2b for only the cells with very sharp tuning and most robust and also sharpest average orientation preference, as these cells will be less susceptible to residual neuropil contamination (e.g. estimated using confidence intervals and shuffle distributions).

Authors' response:

We thank the reviewers for their comments and suggestions.

We were pleased that Reviewers #1 and #2 were satisfied with the revised version of the manuscript.

We felt encouraged that Reviewer #3 found the manuscript much improved, but s/he also offered a number of specific suggestions for consideration.

We felt many of the proposals were good ideas, and implemented nearly all the analyses suggested by Reviewer #3 in this revised version.

This new content is presented in a new sub-section of the Results and along with new **Fig 4** and **Fig 5**.

Briefly, these new data and analyses include:

- Histograms showing the distribution of kurtosis for all inhibitory cell sub-types (PV, SOM and VIP) (**Fig 4 c,d**).
- A demonstration that clustering is still observed after leaving out cells with low kurtosis (putative inhibitory neurons of any class) (**Fig 4e**).
- A demonstration that clustering is still observed when only cells with sharp tuning are considered (**Fig 5a**).
- A demonstration that clustering is still present after leaving out small ROIs that may represent dendritic branches (**Fig 5b**) that, despite our care during the segmentation process, could have been inadvertently included in our dataset.

We have also made additional improvements to the presentation by adding a better illustration and description of how the calcium signals are processed (**Fig 1c**).

Finally, we included some missing references pointed out by Reviewer #3 and discussed them in the context of our findings.

We hope the new content and the replies below address all concerns satisfactorily.

A point-by-point reply to the comments follows.

Reviewers' comments: Reviewer #1 (Remarks to the Author): The authors have addressed all of my concerns.

Thank you. We were glad all concerns were addressed.

I have just two minor suggestions for further edits:

Delete "from": "Individual cells were segmented from the data (Fig 1c) and their spiking inferred from by non-negative deconvolution of the calcium signals..."
Done.

"The finding that tuning is spatially clustered in mouse V1 raises a natural question - do mice have cortical maps?" This statement should be qualified. There is no doubt that mice have strong retinotopic maps. Perhaps change to "- do mice have feature-based cortical maps?"

Done.

Reviewer #2 (Remarks to the Author):

The authors have substantially improved the text and description of their data. This is an important study that will have a large impact on the field. I have only a few comments that I believe will help others realize the importance of this work.

Thank you for this positive assessment.

Issues:

1) It would be helpful to **insert supplementary figure 1 into the manuscript**. It is an important figure and it would be much better if supplementary figure 1 were placed between the current figures 2 and 3.

Following the reviewer's suggestion we have now included one of these panels as **Fig 2c**.

2) The procedure for extracting spikes from the Ca^{++} signals is an important methodological consideration. Supplementary figure 2 shows the calcium waveforms, but it would be nice to show the extracted spikes as well. I do not find the extracted spike histogram in figure 1d as very informative as we do not know the underlying calcium signals that they are based upon.

Following the reviewer's recommendation we have now included examples of the raw traces, neuropil and inferred spikes in **Fig 1c**. The accompanying text and methods were expanded to explain the procedure we use in more detail.

3) If no spike extraction were performed, does the same relationship between similarity and distance hold?

Yes, a very similar relationship holds. The kernels just become spread in time. However, note the concern of Reviewer #3 that processing the raw signals by themselves may include a contribution of the neuropil that may artificially induce clustering.

4) Why is there a different asymptote in figure 2 and figure 3?

Because they are different datasets. In any one individual field, the asymptote will effectively measure the degree of bias in the population as a whole (for example a bias towards horizontal orientations). Different imaging fields show different amounts of bias and, therefore, have different asymptotes.

5) The depth relationship shown in figure 3 is very interesting. I wonder if it might be helpful to add a panel that separates the depths by color, but with an x-axis that reflects the actual distance of between the cell bodies instead of just the cortical distance. If depth distances are the same as lateral distances then all the curves should overlap. I think this might strengthen the columnar argument.

Thank you for this suggestion. A precise comparison between horizontal and vertical clustering would require us to make sure the plane of imaging is nearly perfect parallel to the cortex and collect more data across different optical planes. We are now in the process of repeating these experiments using volumetric imaging with an electrically tuned lens. The collected data will allow us to determine more precisely any relative tilt between the objective axis and the cortical surface that could be corrected and provide a comparison of the space constants in the vertical and horizontal dimensions (or test if they are the same, as proposed by the Reviewer).

Reviewer #3 (Remarks to the Author):

The manuscript is improved from the original submission to Nature, and the new data is useful. Several concerns remain which, if addressed thoroughly, would strengthen the manuscript and its impact on the field.

Thank you. As described below we have implemented most of the proposed analyses and we hope this addresses the Reviewer's major concerns.

We reply only to the second round of the Reviewer's comments which appear below in red.

Previous comments and rebuttal are below, with new comments beginning with --

The authors investigated whether, in mouse primary visual cortex, there is evidence for spatial clustering of neurons that share similar visual orientation and/or spatial frequency tuning. The premise is that previous studies claim that, in contrast to many carnivores and primates, mice and rats lack any such spatial organization in visual cortex. The manuscript is well written. Advances in this manuscript include use of state-of-the-art two-photon microscope, novel methods for eyetracking, novel means for efficient stimulus delivery and mapping, novel sophisticated analyses for brain motion correction and signal extraction. These methods allowed for a high yield of neurons with robust multidimensional tuning curves. However, several considerations limit enthusiasm for publishing this work in Nature:

1) The level of clustering reported is nevertheless quite weak. Whether or not the organization is entirely salt-and-pepper, a change in tuning similarity from .3 to .38 (Fig. 3) for neurons <50 um apart is quite small, and far from many people's conception of a 'minicolumn', the term suggested by the authors.

See our reply to Reviewer #1, point #2. We agree that the degree of clustering is much smaller than seen in higher mammals and we have modified the text to make this clear. The term "minicolumn" was

adopted from the Adams and Horton's review article (which we cite), but the term itself appears to go back to Lorente de No. This modest level of clustering, if real, may not substantially change other scientists' views of the similarity or differences visual cortex in rodents vs., say, cats, where orientation clustering is qualitatively more homogeneous. The data allow us to reject a strict salt-and-pepper organization, and we think the result is highly relevant to the ongoing discussion of cortical organization in different species. Though the degree clustering is weaker than seen in higher mammals, the fact that there is a demonstrable effect opens the possibility that similar mechanisms are at play in different species.

--- It is unclear why the authors still do not cite and discuss Runyan and Sur, 2013 (mentioned below in initial comments). This paper states: "The orientation scatter surrounding both RFP_ and RFP_ neurons increased with distance, being significantly greater at 240 um than at 60 or 90 um ($p < 0.05$ comparing scatter of RFP_ or RFP- neurons at 60 and 90 um with 240 um), reflecting the existence of local homogeneities in the orientation map that dissipate at larger distance scales."

We apologize for the omission. It appears we cited the wrong study. We have corrected this to provide the correct citation to the clustering of PV+ cells in preferred orientation and mention its relation to the present findings in the discussion.

--- It appears that, if the goal of the current paper is to reject strict salt and pepper organization, that the above finding in RFP- neurons may already begin to do that. However, if concerns regarding contamination (see below) are addressed, the current data would nevertheless be quite helpful in consolidating and better quantifying the above finding, as the current methods are more efficient and include more neurons from many more experiments.

Our aim was to test the hypothesis of a salt-and-pepper map (we did not know what we would get, nor was our initial goal to reject the hypothesis). We agree the clustering of PV cells is consistent with our findings and we now cite the paper in the Discussion.

2) Inhibitory interneurons could account for the majority of the effect observed: The authors did not discriminate excitatory from inhibitory neurons. Further, the authors used AAV1-synGCaMP6. AAV1-syn-XX is known to express in inhibitory neurons as well as excitatory neurons, and to show a bias towards PV+ inhibitory neurons (Callaway and colleagues). Given that inhibitory neurons show broad tuning but still often possess orientation and spatial frequency preferences, it's likely that at least 15-20% of neurons included are inhibitory neurons, with most of these being PV+ interneurons. Many studies have shown that interneurons likely pool the responses of nearby excitatory neurons. Recently, Runyan and Sur (2013, J. Neurosci, Fig. 5E) showed that the orientation preference of PV+ interneurons is more similar to the preferences of non-PV neurons (mostly excitatory) located <60 um away from the PV+ neuron cell body, and that this similarity in preference is weaker when considering neurons <240 um away. From this, one can infer that the PV+ neurons are more likely than chance to match the tuning of their nearest neighbors, in which case lumping PV+ and PV- neurons in the current study could very well help explain the modest local clustering result in the current manuscript. This same argument can be applied to the data the authors reanalyzed (current manuscript, Fig. 4) from a previous paper (Ohki et al), which also did not distinguish interneurons from excitatory neurons. Runyan and Sur also show a very weak local clustering of orientation preference (<5%) amongst PV-putative excitatory neurons. This can either be seen as prior evidence that even excitatory neurons have already been shown to exhibit some local clustering. Alternatively, these data could be seen as prior evidence that other studies find only trivially low levels of orientation clustering when excluding

PV+ neurons. Similar to Runyan and Sur, a subsequent study also showed evidence of a bias in tuning (in this case, in ocular dominance tuning as well as disparity tuning) of PV+ neurons to nearby PV- neuron tuning, at the 50 μ m scale in mouse visual cortex (Scholl .. Zemelman, Neuron 2014, Fig. 4). Further, Scholl et al. saw little or no spatial clustering for these other features in mouse V1 when considering PV- neurons (mostly excitatory), further suggesting that V1 in rodent and some carnivores/primates may be qualitatively different in this regard.

This is a key point that the Nature editor asked us to consider as well before resubmission. We now include new data quantifying the level of contamination expected from PV+ neurons; GCaMP6f signals simultaneously measured from pyramidal neurons and PV+ neurons, the latter identified by their expression of tdTomato (n=4 mice), and compared their signal statistics. The signals generated by PV+ cells are easily distinguished from those generated from pyramidal cells, and easily separated by the kurtosis of the responses, as is clear in our new Supplementary Figure 2. Pyramidal neurons generate large and transient calcium spikes that result in very large kurtosis values (Supplementary Fig 2b, top). In contrast PV+ neurons generate kurtosis values that are only slightly above 3 (corresponding to a Gaussian distribution) (Supplementary Fig 2b, bottom and Supplementary Fig 2c). Moreover, the density of PV+ cells is much lower than those of pyramidal neurons. Even if we overestimate the relative ratio of PV+:PYR at 1:4, this makes the probability of inclusion of a PV+ cell in our dataset at $p=0.0025$. Then, the probability that a pair of cells selected in the analysis will include at least one PV cell, will be 0.5% at the most. As additional confirmation, a re-analysis of the data where we reject any cell with kurtosis < 10 (a threshold never exceeded by a PV+ cell in the control experiments) yields effectively the same results reported here.

--- The discussion in Supp fig 2 is quite useful, though it remains unclear from the main Methods what criteria were used. What does 'high kurtosis' mean? Can you please define exactly what the kurtosis cutoff is? Was it $k > 7.35$? This is discussed in text in terms of back-of-the-envelope calculations, but not specified in methods as the cutoff that was actually used.

--- While the kurtosis arguments are useful, the estimate of 1:4 PV cells doesn't fully address the issue of other GABAergic interneurons, all of which are broadly tuned and should at least be discussed.

--- Further, due to overall broader tuning, PV cells are more likely to be driven by one of the chosen SFs and TFs and orientations and stimulus sizes used than a typical non-PV cell, thus the driven subset of cells in the sample will likely be enriched for PV cells, in cases where comprehensive mapping of the entire stimulus space is not performed. These considerations, while unlikely to strongly affect the outcome, merit further discussion and more detailed methods.

We now provide data from a new set of animals to compare the kurtosis of all inhibitory cell types (**Fig 4c,d**) to those of excitatory neurons. We show that pyramidal and inhibitory cells can be largely segregated based on the kurtosis of their signals. We then demonstrate that clustering of tuning remains present after removing the signals with low kurtosis from our dataset (**Fig 4e**). The threshold for kurtosis after considering all cell types was set at 15, as now stated in the text and Figure.

--- Surprisingly, the Runyan and Sur 2013 paper showing weak clustering among PV- neurons is still not included. A citation to Runyan 2010 is provided at the start of the new section, which would appear to be the wrong citation.

--- Also, Scholl / Zemelman reference is still not included, even though this work directly discusses the issue of clustering in PV+ and in PV- neurons in mouse V1.

These two studies are now included in the Discussion.

3) Several technical issues make it difficult to properly evaluate the level of clustering, given the reagents used and the current explanation of the analysis methods. First, the authors use GCaMP6, which is known to drive strong responses in apical dendritic branches. Given the imaging objective used and any residual brain motion, if even a few of these branches were included as spurious cell bodies, they could contribute to local clustering. Nearby proximal dendritic branches from the same neuron will likely have similar tuning and thus could bias the clustering effect, even across planes 40-120 um apart, since pyramidal cells are radially organized. As described in the Methods section images were stabilized before signals were extracted. The non-rigid alignment process results in residual motion that is a small fraction of the ROI representing the cell body, and mitigates artifacts that result from motion in the (x,y) plane. Typical examples of such compensation in our setup and how they compare to rigid alignment, together with the code we use, can be seen here: <http://xcorr.net/2014/08/02/non-rigid-deformation-for-calcium-imaging-frame-alignment/>

--- Response to point 3 addresses motion correction in x-y (not in z).

We do not see any significant motion in z in our setup.

--- This response is still incomplete, and doesn't address possibility of multiple dendritic branches being counted. Are results weaker if the smallest diameter ROIs (e.g. possible dendrites from same neuron) are excluded?

Following the reviewer's recommendation, we now provide such an analysis in **Fig 5b**

A second challenge in parsing the data is the unconventional, albeit interesting, data analysis methods used. Dense infection with GCaMP6 is known to result in huge neuropil signals (Chen .. Svoboda). If not removed perfectly for each neuron, these signals will contaminate the cell body of interest. Neuropil itself can exhibit local tuning biases, further compounding the issues. Here, there is no way to assess if the authors ensured that neuropil contamination was fully eliminated - an important prerequisite for these particular claims. This is even more problematic given the large point-spread function of the imaging objective employed.

The contributions from the neuropil were subtracted by estimating a robust linear predictor of the signal based on the surrounding neuropil (excluding other nearby cells) and then subtracting it from the ROI signal. The scatterplot of neuropil vs cell's ROI signal typically shows that the baseline of the cell body ROI is linearly related to the neuropil (both signals z-scored):

--- Regarding neuropil - even if the fit removed 90% of neuropil contamination, the residual 10% might be enough to drive the very weak signals shown, particularly in poorly tuned neurons. Neuropil contamination is a serious issue with GCaMP6, it should be taken seriously, particularly when analyzing very subtle effects. Were the clustering finding to be true despite residual neuropil contamination, neuropil contamination would still lead to an overestimate of this already small effect.

--- Additional analyses could include comparisons of ori preference of neuropil vs cells, reanalysis of

Fig 2b for only the cells with very sharp tuning and most robust and also sharpest average orientation preference, as these cells will be less susceptible to residual neuropil contamination (e.g. estimated using confidence intervals and shuffle distributions).

We followed one of the suggestions here and re-analyzed the data by focusing on cells with well-tuned receptive fields, which we selected by requiring the peak of the kernel to be located at a spatial frequency larger than 0.025 cycles/deg. A re-analysis of these data shows that clustering remains even while concentrating on this group of sharply tuned cells (**Fig 5a**).

In addition this analysis, we would like to point out that our methods are such that the trace representing the spike inference is essentially zero in segments that contain no spikes (see **Fig 1c**). This effectively “squashes” any small background activity that may be shared across cells from the neuropil making any potential contribution extremely unlikely.

Reviewers' comments:

Reviewer #2 (Remarks to the Author):

No further comments.

Reviewer #3 (Remarks to the Author):

The authors have done a very nice job adding additional analyses that now make the argument quite strongly that there is a small amount of clustering in tuning of nearby pyramidal neurons in V1. Only Two small Discussion-related issues remain:

1) First, while the authors now cite Runyan and Sur, J Neurosci 2013, they only discuss the existence of clustering in PV+ neurons, while my comments also point to the finding they claim of clustering (which is admittedly much weaker and more questionable) in RFP-neurons, ie PV-neurons. In the 2013 paper, Runyan and Sur write:

The orientation scatter surrounding both RFP-and RFP+ neurons increased with distance, being significantly greater at 240 um than at 60 or 90 um ($p < 0.05$ comparing scatter of RFP+ or RFP-neurons at 60 and 90 um with 240 um), reflecting the existence of local homogeneities in the orientation map that dissipate at larger distance scales.

The current manuscript now shows that all interneuron subtypes in V1 have low kurtosis in their calcium-evoked activity. This allows them to make a stronger claim that the statements they make regard excitatory neurons, and not just PV-neurons (which could in Runyan and Sur's case be due to clustering among SOM or VIP neurons).

The previous reported (weak) finding of clustering in 'RFP-neurons' should be discussed and the valuable improvements to interpretation afforded by the current manuscript should be illustrated.

2) The author's analyses and improved discussion of the low likelihood of neuropil contribution is useful. The authors now write "Finally, the probability of spiking is inferred by non-negative deconvolution (see Methods for details). The result is a trace that is nearly identical to zero in regions devoid of spiking activity (red trace), further ameliorating small contributions"

- the last part of this sentence requires clarification ('further arguing against a contribution from neuropil?'), and also is missing a period.

Authors' response:

Below are point-by-point replies to the reviewers' comments.

REVIEWERS' COMMENTS:

Reviewer #2 (Remarks to the Author):

No further comments.

Thank you.

Reviewer #3 (Remarks to the Author):

The authors have done a very nice job adding additional analyses that now make the argument quite strongly that there is a small amount of clustering in tuning of nearby pyramidal neurons in V1. Only Two small Discussion-related issues remain:

Thank you for your previous suggestions -- they were useful.

1) First, while the authors now cite Runyan and Sur, J Neurosci 2013, they only discuss the existence of clustering in PV+ neurons, while my comments also point to the finding they claim of clustering (which is admittedly much weaker and more questionable) in RFP- neurons, ie PV-neurons. In the 2013 paper, Runyan and Sur write:

The orientation scatter surrounding both RFP- and RFP+ neurons increased with distance, being significantly greater at 240 um than at 60 or 90 um ($p < 0.05$ comparing scatter of RFP+ or RFP-neurons at 60 and 90 um with 240 um), reflecting the existence of local homogeneities in the orientation map that dissipate at larger distance scales.

The current manuscript now shows that all interneuron subtypes in V1 have low kurtosis in their calcium-evoked activity. This allows them to make a stronger claim that the statements they make regard excitatory neurons, and not just PV- neurons (which could in Runyan and Sur's case be due to clustering among SOM or VIP neurons).

The previous reported (weak) finding of clustering in 'RFP- neurons' should be discussed and the valuable improvements to interpretation afforded by the current manuscript should be illustrated.

Thank you. We now indicate that the data presented in Fig 4d refines the previous result described

in Runyan and Sur by allowing us to exclude all inhibitory interneurons from the analysis.

2) The author's analyses and improved discussion of the low likelihood of neuropil contribution is useful. The authors now write "Finally, the probability of spiking is inferred by non-negative deconvolution (see Methods for details). The result is a trace that is nearly identical to zero in regions devoid of spiking activity (red trace), further ameliorating small contributions" - the last part of this sentence requires clarification ('further arguing against a contribution from neuropil?'), and also is missing a period.

Thank you. This sentence was inadvertently truncated for some reason. It has been restored and clarified.